# Examine Race/Ethnicity Disparities in Perception, Intention, and Screening of Dementia in a Community Setting: Scoping Review

**DOI:** 10.3390/ijerph19148865

**Published:** 2022-07-21

**Authors:** SangA Lee, Deogwoon Kim, Haeok Lee

**Affiliations:** Nursing Department, Robert and Donna Manning College of Nursing and Health Sciences, University of Massachusetts Boston, Boston, MA 02125, USA; sanga.lee001@umb.edu (S.L.); deogwoon.kim001@umb.edu (D.K.)

**Keywords:** older adults, Alzheimer’s disease, dementia, screening, race, ethnicity, health disparities

## Abstract

Background: Delayed detection and diagnosis of Alzheimer’s Disease and related dementia (ADRD) can lead to suboptimal care and socioeconomic burdens on individuals, families, and communities. Our objective is to investigate dementia screening behavior focusing on minority older populations and assess whether there are ethnic differences in ADRD screening behavior. Methods: The scoping review method was utilized to examine ADRD screening behavior and contributing factors for missed and delayed screening/diagnosis focusing on race/ethnicity. Results: 2288 papers were identified, of which 21 met the inclusion criteria. We identified six dimensions of ADRD screening behavior: Noticing Symptoms, Recognizing a problem, Accepting Screen, Intending Screen, Action, and Integrating with time. Final findings were organized into study race/ethnicity, theoretical background, the methods of quantitative and qualitative studies, description and measures of ADRD screening behavior, and racial/ethnic differences in ADRD screening behavior. Conclusions: A trend in ethnic disparities in screening for ADRD was observed. Our findings point to the fact that there is a scarcity of studies focusing on describing ethnic-specific ADRD screening behavior as well as a lack of those examining the impact of ethnicity on ADRD screening behavior, especially studies where Asian Americans are almost invisible.

## 1. Introduction

According to an annual report of the Alzheimer’s Association, more than six million Americans live with Alzheimer’s disease and related dementia (ADRD). Between 2000 and 2019, one in three seniors died from ADRD, while the number of deaths from heart disease decreased by 7.3% [1]. Currently, more than 55 million people live with dementia globally, and there are nearly 10 million new cases every year. The magnitude of health economics and the human and social impact on individuals living with dementia, their families, communities, and society are drastically increasing. It is estimated that the global cost of dementia could grow to US$2 trillion by 2030.

Although there are currently no specific treatments to block the progression of cognitive decline in ADRD, an early diagnosis helps patients and families to plan for their future care and treatment. However, dementia is substantially underdiagnosed by clinicians and underreported by patients and families in the U.S., and studies have found that 40 percent to more than half of patients with dementia had not received a clinical cognitive evaluation by a physician [2,3,4]. Moreover, according to Healthy People 2020, baseline data show that, even among those diagnosed with dementia, nearly two-thirds of them or their caregivers were aware of the diagnosis [5]. Studies show that racial/ethnic disparities in misdiagnosis and the problem of underdiagnosis are even more pronounced in underserved populations and those with low socioeconomic status (SES) [6,7,8,9,10,11]. In the same vein, evidence points to the fact that, at first dementia diagnosis, racial/ethnic minorities are more impaired and show more severe clinical symptoms, which may suggest that diagnosis occurs at a later stage of the disease for these groups [12,13]. The older population is becoming more racially and ethnically diverse. Between 2014 and 2060, the share of the older non-Hispanic white population is projected to drop by 24 percentage points, from 78.3 percent to 54.6 percent, in the U.S.; therefore, a higher percentage of people with ADRD will be minorities [14]. The literature supports the need for studies exploring and explaining the impact of the ethnoracial factors on ADRD, not only genetic and biological factors, but also sociocultural factors related to the timing of diagnosis, lay symptom recognition, lay diagnosis and course of AD between different ethnocidal groups [15,16].

Differences in dementia screening behavior from both healthcare professionals and patients or patient families can be expected as a function of the ethnoracial factors because the ethnoracial groups are characterized by distinct social and behavioral practices [16,17]. Several factors related to dementia screening and diagnosis, including the individual level of health belief, sociocultural norms, and access to healthcare services among elderly minority populations, pose special challenges in dementia diagnosis and treatment [16,17,18,19,20,21]. However, there is scarce information on the extent of understanding dementia screening behavior among ethnic minorities. Understanding dementia screening behavior and race/ethnicity as influencing the proportion of detected and undetected dementia is important for improving dementia screening/diagnosing behaviors and moving to equity of ADRD management.

This review aims to investigate dementia screening behavior among minority older populations and whether there are any racial/ethnic differences in screening behaviors. Thus, we only reviewed studies that addressed ADRD screening/diagnosing behavior among ethnic/racial minorities in North America versus reports of screening/diagnosis rates.

## 2. Materials and Methods

We adopted the Preferred Reporting Items for Systematic Reviews and Meta-analyses extension for Scoping Reviews (PRISMA-ScR) [22]. A scoping review uses systematic and rigorous methods and summarizes the findings from the body of knowledge that is heterogeneous in methods including both quantitative and qualitative designs or disciplines [22,23,24,25,26,27]. The difference from a systematic review is that a scoping review does not have a specific research question, but rather seeks to provide a broad overview of the available literature and identify gaps in the literature [24,26]. Our protocol was drafted using both PRISMA-ScR and Arksey and O’Malley’s scoping review guidelines [22,23,24,25]. No ethical approval is required for this type of study. 

### 2.1. Search Strategy and Study Selection

The search strategy included databases, search terms, and inclusion/exclusion criteria. Three databases (PubMed, Cumulative Index to Nursing and Allied Health Literature (CINAHL), and PsycINFO) were used to search for the most relevant literature. Search terms to find published studies that reported on factors associated with dementia screening behaviors were determined through several discussions with a university librarian: “dementia screening”, “Alzheimer’s disease screening”, “barriers”, “delayed”, “facilitators”, factors”, “pathway”, “race”, “ethnicity”, “minorities”, “Asian American”, “Black/African American”, “Hispanic”, “Latino/a”, and “Native American”, combined with the Boolean operator “AND/OR.” The publication year of the literature was restricted from 2000 onwards because the number of deaths from Alzheimer’s disease (AD) has increased by more than 145% from 2000 to 2019 in the U.S. [1]. One of the authors (H.L.) who designed this study recommended several studies of examples that might be included in the literature. An additional database search for more studies targeting Asian Americans was conducted. For this search, the university librarian designed additional search terms as follows: “Chinese American”, “Filipino American”, “Asian Indian American”, “Korean American”, “Vietnamese American”, and “Japanese American”, which are the five largest Asian American subgroups. 

Prior to the initiation of screening, duplicate records were removed. After duplicates were removed, articles retrieved from the databases and the author’s recommendations were screened for their titles, abstracts, and full texts to determine their eligibility. Only empirical articles with an English abstract were included. Since the aim of the scoping review is to provide a broad overview of relevant studies, we considered all types of research design. We applied the following inclusion criteria at two stages of study selection (screening by title and abstract, and then full text). Inclusion criteria involved: (a) use of primary data reporting factors relative to ADRD screening behaviors, (b) conducted in North America (the U.S. or Canada), (c) covering the older adult population, (d) published in English, and (e) full text available. Excluded criteria were studies: (a) focused on countries outside North America, (b) did not measure ADRD screening behavior, and (c) did not report any data for minority populations as this study sought to examine dementia screening behavior among the ethnic minority older adult population and there are any disparities in dementia screening. 

To increase consistency among reviewers, three researchers (H.L., S.L., D.K.) independently performed screening by assigning studies to “include,” “exclude,” or “not sure” categories and discussed the results. Discrepancies among researchers were resolved through discussion, and the screening and data extraction guidance was revised.

Lastly, reference lists of included studies were hand searched and screened to ensure the complete inclusion of relevant studies. Final decisions about the inclusion of studies were made upon the discussion and agreement between researchers. Figure 1 presents the process diagram indicating the search process.

### 2.2. Data Extraction

Each full text of the included studies was critically assessed to extract data. Data were extracted using a standardized data extraction table that the research team developed, including (a) characteristics such as design, research setting, and ethnicity of participants, (b) theory or framework, (c) dementia screening behaviors, (d) measures, and (e) major findings and discussion. Three reviewers jointly developed a data-charting form to determine which variable to extract. The two reviewers (S.L. and D.K.) independently charted the data, discussed the results, and continuously updated the data charting in an interactive process. These data were extracted by two researchers (S.L. and D.K.) and then examined and agreed upon by the PI (H.L.) to ensure validity.

### 2.3. Collecting and Summarization

The scoping review findings of the literature are documented in Table 1, Table 2 and Table 3 in the narrative text. A descriptive-analytical method was utilized to present a narrative account of the existing literature. Data-charting tables were developed to sort the extracted data and included charting of key features. The tables include the geographical distribution of studies, research methods, dimensions and measures of screening behavior, race/ethnicity of participants, and major findings; the studies are listed in the chronological order of publication date, from the earliest to the most recent publication.

## 3. Results

### 3.1. Search Results

The final articles about ADRD screening or diagnosis behavior, and factors associated with screening behavior were analyzed. The final articles were organized into study locations, designs, sampling, race/ethnicity and language, and major findings of dementia screening behavior (Table 1). In order to provide structure and meaning to the results, we created thematic frameworks [23]. Three frameworks were created including study characteristics, dimensions of ADRD screening behaviors, and ADRD screening behaviors by race/ethnicity.

### 3.2. Study Designs

Table 1 presents the quantitative, mixed-methods, and qualitative studies. Among the quantitative studies, five studies used the existing national health survey data or Medicare/Medicaid data. No experimental studies were included. As can be seen by reviewing Table 1, most studies were cross-sectional surveys that were conducted in the U.S. Various qualitative methods were used, including open-ended interviews, focus groups, grounded theory, semi-structured and in-depth interviews, and ethnography. The studies took place in the U.S. (*n* = 17) and Canada (*n* = 4). Race/ethnicities studied were: (1) Whites; (2) Blacks; (3) Hispanics; (4) Asian Americans; (5) Native Americans; and (6) others. Subgroups of Blacks were African Americans and Caribbean Blacks, subgroups of Whites were non-Hispanic Whites and French-speaking Whites, and subgroups of Asian Americans were Chinese, Korean, South Asian, Indian, and Sri Lankan.

### 3.3. Sample Characteristics

The most frequently studied ethnic group was Blacks (*n* = 14), followed by Asians (*n* = 7) and non-white Latinos (*n* = 6) ethnic groups. Two studies covered Native Americans [34,36]. All areas, urban (*n* = 11), suburban (*n* = 4), and rural (*n* = 4), were included. Fourteen papers studied patients or people with and without dementia in their research. Nine papers involved only patients, and five studies involved both patients and family caregivers as dyads. Patients with or without dementia were the largest group of participants in the identified studies. In terms of settings of the studies, 17 studies were conducted in the U.S., and four in Canada. Subjects were recruited from clinics, community-based clinics, or community organizations. All primary quantitative studies recruited the participants in person through the partner agencies’ networks, phone calls, advertising materials (e.g., flyers, newspapers, and social media). Five studies utilized the existing population-based national data about health insurance (e.g., the National Health and Aging Trends Study (NHATS) and California Medicare fee-for-service data) or older adults’ economic, health, and psychosocial information (e.g., the Health and Retirement Study (HRS)). Of the five, three studies utilized a combination of in-person and online recruitment methods such as email [39,40] and social media [33]. Potential participants were recruited by referral from local community centers, physician offices, churches, senior centers, and memory disorder clinics. Only two studies [32,35] included a report of the response rate, and one study [35] included a report of response rates below 7.5%. As this study focused on North America, most studies (*n* = 12) used English, except for seven studies without language information for the recruitment or data collection. Of the 12, five studies reported “English-speaking” as an inclusion criterion. Seven studies used both English and other minorities’ languages, such as Spanish (*n* = 4), Korean (*n* = 1), Chinese (*n* = 1), and Hindi (*n* = 1). Two studies used only the language of their target population, French or Chinese.

### 3.4. Theoretical Trends

Most quantitative studies were conducted without theoretical frameworks. Only two cross-sectional studies [33,36] used theoretical frameworks: a modified version of the Integrated Behavior Model [36], and a conceptual framework were developed based on the attitude theories focusing on stigma beliefs and subjective norms of AD care among Korean Americans [33]. Two qualitative [43,45] and one mixed-methods [39] studies used health belief models or a health-seeking model to explore the perception of dementia screening and to explain the rationale for dementia screening-seeking behaviors among ethnically diverse patients, family caregivers, and stakeholders. According to the Kaleidoscope Model of Health Communication, Garcia and others’ study [47] looked at health communication between HCPs and French-speaking patients, especially people with dementia who reverted to their primary languages among Francophones.

### 3.5. ADRD Screening Behavior Is A Multidimensional Construct

We have found that two terms of AD and dementia are often used interchangeably in the study and practice; however, AD is preferred more than dementia because patients more readily understand the term [30]. Hence, the term ADRD screening behavior will be used in this paper for dementia and AD screening behavior. The literature revealed that ADRD screening behavior is a multidimensional construct including both subjective perception and objective screening action, specifically, dimensions of noticing symptoms, recognizing a problem, accepting screen, intending to screen, action, and integrating with time (Table 2). ADRD screening behavior is integrated with time as it is defined as delayed, timely, or early screenings. Most studies focused on recognition of the problem (*n* = 6), acceptance or intention (*n* = 4) to do screening of ADRD, and actual ADRD screening (*n* = 4). The empirical statements for each dimension extracted from the included studies were summarized and are compared in Table 2.

Though the phenomenon of ADRD screening behavior is on a continuum, and it is not static, most quantitative studies utilized cross-sectional design and typically examined associations between screening behavior and its correlates as static. Qualitative [42,43,44,45,46,47] studies addressed that dementia screening behavior is not static and that it is fluid, and screening pathways which were multiple stages of the screening process were identified by examining and reporting recognition of dementia symptoms, making lay diagnosis or awareness of the problem, intention to screening, and then taking screening examinations. However, some people can move back and forth along the pathways. Koehn and others [45] stated that most Chinese Canadian caregivers described a process of gradually piecing together the rationale to explain their spouses’ or parents’ “puzzling behaviors”, and Leung and others [44] described that none of the Euro-Canadian participants identified any critical event, but rather a gradual memory decline when asked what triggered their desire to seek care. However, once Chinese Canadian caregivers realized that the symptoms and behaviors were “problematic”, they quickly sought additional information, usually from a family physician [45].

As for measurements, there is limited information assessing the multidimensional and time-bounded pheromone of ADRD seeking behavior. There is an absence of a commonly agreed-upon approach to assess ADRD seeking behavior, as shown in Table 2. In general, dementia screening behaviors were measured based on a focus on the types of dimensions of the construct of screening behavior that they were interested in studying (Table 2). Most measures were developed by authors for their studies or revised the existing tools which measured similar concepts [29,31,33,36,39]. For instance, the PRISM-PC questionnaire, which is most often used [30,31,39,40], consists of two separate scales: the patient’s acceptance of the dementia screening scale and the patient’s perceived harms and benefits of the dementia screening scale. The measure of action included both whether ADRD screening/diagnosis was missed or delayed [37], or screened/diagnosed ADRD involved the use of clinical interviews and previously standardized and/or validated assessment criteria [7,34,35,38]. Missed/delayed screening/diagnosis were usually assessed using the proportion of patients or people within the research settings who were considered to have ADRD but who were not diagnosed by their physicians or had no information in their medical records. Five of 17 quantitative studies used the existing population-based data in conjunction with Medicare and Medicaid data [7,32,34,37,38]. Regarding screening behavior integrated with timing, the interval was measured from noticing or recognition of symptoms of dementia either by patients, caregivers, or HCPs to first contacting a physician or actual screening. The term “Delay” is often used to report the intervals that were reported in days or months and categorized into two delays: delay before recognizing the problem of symptoms and delay before patients or caregivers contact physicians [28]. The time interval or delay was measured in months, and reported delay times before recognizing problems of symptoms and before physician consultation was 0 to 84 months and 0.1 to 84 months, respectively [28].

### 3.6. Differences in ADRD Screening Behavior among Ethnicities

We first provide comparative summaries on awareness, intention, and action of screening and diagnosis among minority ethnic groups (Table 3). We then summarize the quantitative reports about African Americans/Blacks, Hispanics/Latinos, and Asian American groups. Several studies either reported on individual ethnic groups, in aggregate or multiple ethnic groups. The majority of studies reported that minority groups had low awareness, intention, and diagnosis, compared to non-Hispanic White reference groups [7,31,37,41]. Differences between minority ethnic groups were more evident from population-based research than from clinic-based or community-based studies (Table 1 and Table 3). For example, Gianattasio et al. [7] reported that Blacks had double the risk of being underdiagnosed compared with Whites in all six observations (RRs = 1.58–2.4) based on the Health Retirement Study (N = 29,775; 2000 to 2010). Similarly, Tsoy et al. (2021) [38] used a large data set of 2013–2015 California Medicare fee-for-services (N = 10,472) and found low ORs for receiving a timely diagnosis among Asians (OR = 0.46; 95% CI = 0.38–0.56), Blacks (OR = 0.73; 95% CI = 0.56–0.94), or Hispanics (OR = 0.62; 95% CI = 0.52–0.72) compared to Whites. Asians (incidence RR = 0.81; 95% CI = 0.74–0.87) also received fewer diagnostic evaluations, and these associations remained significant after adjusting for age, sex, comorbidity burden, neighborhood disadvantage, and rurality.

Among six qualitative studies, four were reported from Canada, of which three reported on Chinese, South Asian, and French-speaking groups [45,46,47]. Most qualitative studies explored experiences in the perception of noticing symptoms and how recognizing symptoms as problematic translated into making a lay diagnosis to seek medical attention from patients, family, and stakeholders. Though there are multiple pathways of the screening process from noticing symptoms to having been screened, no ethnic-specific pathways were reported. Rather, there is a general trend that, at the first notice of symptoms, most participants, regardless of their ethnicity and whether they are patients or family members, considered it as a normal aging process or temporary episodes. However, among Chinese, South Asian, and French-Canadian subgroups, themes of cultural communication emerged and were linked with delayed diagnosis. The following quotes illustrate the point:

“He recalls that his mother’s doctor, who practices in Chinatown, said, ‘don’t expect too much… it is like a machine. After a long period of time, its parts will certainly be off or fall apart.’ The son denies that his mother has ever received a formal diagnosis of AD or dementia. He wishes that the doctor would ‘explain more’ and emphasized language barriers as a significant problem in getting care for his mother.”[42] (p. 140)

In another study, they pointed to the fact that they would like to have physicians who speak the same ethnic language even though they speak English. “He (patient) lost a lot of his English…we had settled down for good…, there was the possibility of getting a doctor in French” [47] (p. 972). They were waiting for a physician who could communicate in French, which is one of the reasons for the delay in screening. In addition, in the Asian, Chinese subgroup, a stigma theme emerged: “I think my family did not have this before. I thought that when people got old, they would be forgetful, talked silly and sometimes insane. It is not a disease, but a natural course of life…because in old age, it is like that -it should be crazy (Shu*)” [45] (p. 49).

## 4. Discussion

With no effective treatment to prevent or modify the disease course, AD is an immense burden on our economy, patients, and caregivers; hence, early screening of ADRD based on early recognition of the problem of symptoms and caregivers’ concerns is critical to identify reversible etiologies and reduce patient and caregiver burden. This paper has performed a scope literature review of ADRD screening behavior focusing on ethnic minorities in North America since the racial/ethnicity factors that contribute to screening behavior are not well understood. The results point to the fact that there is a scarcity of studies focusing on describing ethnic-specific ADRD screening behavior as well as a lack of those examining the impact of ethnicities on ADRD screening behavior; especially, studies with Asian Americans are almost invisible. Although Asian Americans are the fastest-growing population in the U.S., ADRD research on Asian Americans is limited. It is reported that clinical research focused on Asian Americans and funded by the NIH only comprised 0.17% of its total budget based on 529 projects [48]. Although the goals of the 2012 National Plan to Address AD include strategies to diversify outreach in AD research [49], as observed in this review, investment and data for AD research on Asian Americans remain insufficient. As the incidence of ADRD increases significantly with age, especially among racial and ethnic minority groups, the fastest-growing population in the U.S., we believe our findings are very timely to exhort health policymakers and the scientific community to conduct more studies to enhance our understanding of this underserved and understudied population.

The most common first signs and symptoms noticed by the patient or others were memory loss, repeated questions, or word-finding problems, and the common cues to a willingness to screen were concerns about safety for patients, others, and the environment. These results are in agreement with two previous systematic literature reviews of non-ethnic group selected studies [17,20], suggesting that the association of concern about the safety of patients with prompting them to seek screening is generalizable. A lack of awareness, misunderstanding, access to healthcare resources, and stigma around ADRD combined with few treatment options are barriers to ADRD screening behavior. However, this conclusion could be incomplete due to the relatively small numbers of papers with heterogeneous designs with diverse racial/ethnic groups.

More racial and ethnic minorities were not aware of their cognitive symptoms, not talking to their HCP about their problems, were not being diagnosed, or among patients who were diagnosed, their family members were not informed by HCPs. These results align with previous findings from systemic literature reviews [50,51]. Both patients and caregivers emphasized that there were accumulated symptoms including memory problems, repeated questions, and a decline in judgment including handling checks and bank accounts, which implies that those symptoms could not be related to a single event. Previous literature has pointed out that non-ethnically matched HCPs’ misinterpretations of cultural behaviors might cause delayed diagnosis or misdiagnosis [52]. However, we did not find a positive impact of ethnically matched patients and HCPs on timely screening and cultural communication with caregivers from qualitative studies, which might be due to diversity within the racial/ethnic populations in North America. However, this finding suggests that HCPs will encounter more culturally diverse patients and there is no textbook for this encounter, but there is a need to study qualitatively and quantitatively racially and culturally diverse groups.

### Strengths and Limitations

We included many search terms related to race/ethnicity and included both quantitative and qualitative methods. However, we only included studies conducted in North America and reviewed articles studying racial/ethnic minorities. The fact that only a few studies investigated Asian Americans in our report might be due to our decision to include only peer-reviewed articles. This review did not assess the methodological quality of the included studies as we reviewed both qualitative and quantitative methodologies. However, we think that this body of systematic scoping reviews provides important information that ADRD screening behavior is a multidimensional phenomenon, and it is not static but changes along the continuum of racial/ethnic minorities and that there is a lack of race/ethnicity-specific research, especially among immigrant minority populations. Health care and health research related to ADRD for immigrants is particularly challenging with the triple factors of low health literacy, cultural differences, and limited English proficiency [53], which would act as barriers to an effective screening of ADRD. As well, their impact on recruitment and how research is conducted should be addressed in health policy and research priority settings.

ADRD screening behavior in diverse populations may also be influenced by other sociodemographic factors including educational level, healthcare access, and relationships with healthcare professionals [26,32,33,35,40,42,47,53,54,55,56,57,58,59]. These factors might moderate the relationship between race/ethnicity and ADRD screening seeking behavior. For example, differences in educational attainment may account for the race-specific variability in screening seeking behavior [26,33,40]. Despite near-universal healthcare coverage through Medicare for seniors in the U.S, there are consistent reports of financial barriers to healthcare access, especially among racial/ethnic minority elders in the U.S. compared to Canada and other European countries [53,54]. In addition, racial/ethnic minorities may also have different perceptions and attitudes toward ADRD screening procedures than their White counterparts [55,56,57,58,59]. There exists evidence that Black adults mistrust the healthcare system and White healthcare providers because of a history of mistreatment as well as consistent racial disparities in health and health care [58,59,60]. Mistrust of healthcare providers or the healthcare system and being unable to communicate clearly with healthcare providers about their symptoms of dementia which are culturally bounded may lead to a great impact on ADRD screening seeking behavior. Exploring these sociocultural factors related to ADRD screening seeking behavior across cultures is a promising avenue for future research and we should pay more attention to addressing this problem.

Given that this review focused on studies conducted in the U.S. (English) and Canada (English/French), the language barrier is considered one of the main barriers to ADRD screening behavior in these monolingual countries. However, this observation might differ from countries where multiple languages are commonly spoken (e.g., in several European countries). Thus, our focus on the healthcare context in U.S. and Canada might limit the generalizability of our results.

## 5. Conclusions

A trend in ethnic disparities in screening for ADRD was observed. Timely detection of dementia is equally important as well as knowing the neuropathophysiological diagnosis. However, our findings point to the fact that there is a scarcity of studies focusing on describing ethnic-specific ADRD screening behavior as well as a lack of those examining the impact of ethnicity on ADRD screening behavior, especially studies where Asian Americans are almost invisible. We hope this article will ignite further inquiries to address how and what race–ethnic-related sociocultural factors such as education, healthcare access, health communication, and trusted healthcare providers have an influence on ADRD screening behavior and will impact health researchers, health policy makers, health practitioners, and stakeholders to be aware and to address the problems of ADRD screening seeking behaviors, especially among underserved and understudied racial–ethnic minority populations.

## Figures and Tables

**Figure 1 ijerph-19-08865-f001:**
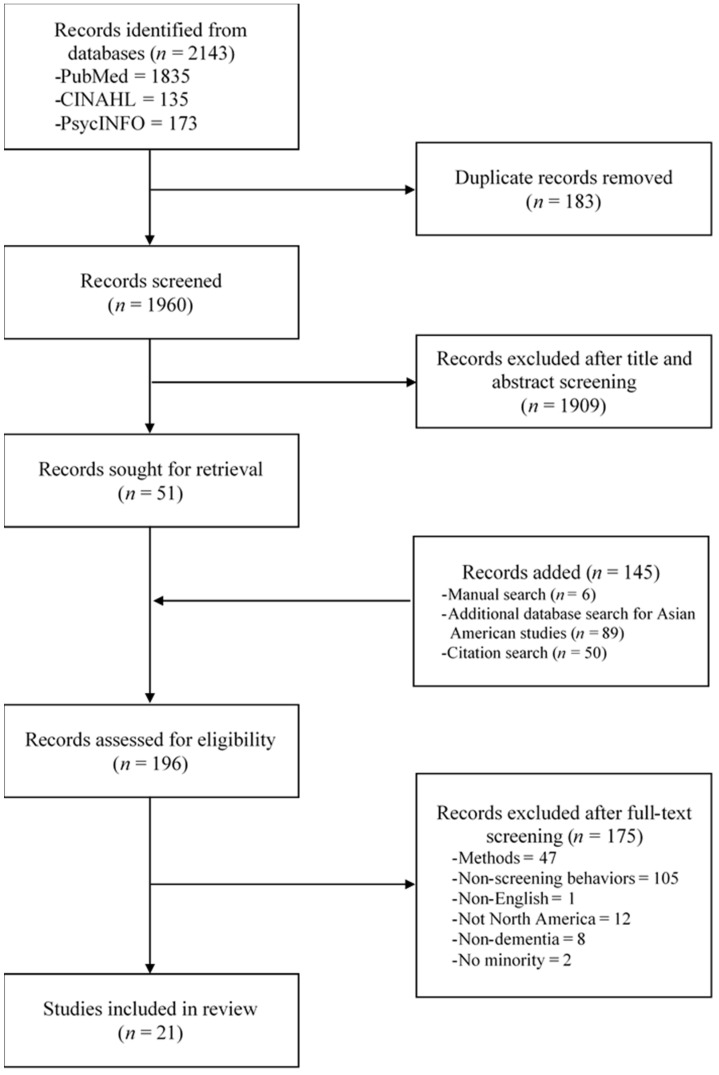
Study Flowgram.

**Table 1 ijerph-19-08865-t001:** Study characteristics and findings.

References	Study Location/ Recruitment Sites/ Age Range	Design/Population/ Sample Size	Ethnicity/Race/ Language	Key Findings
Clark et al. (2005) [28]	U.S. Community-based clinics (urban and suburban)	Cohort study Patients with probable AD and primary family caregivers *n* = 79 units	African Americans	Time from noticing first AD signs to recognition No difference urban vs. suburban (Median 9 months (range 1–84) vs. 6 months (0–72)) Time from recognition to physician consultation: No difference urban vs. suburban (Median 3 months (0.2–84) vs. 2 months (0.1–48)) Longer delay in recognition associated with longer delay in physician consultation
Holsinger et al. (2011) [29]	U.S. 2 sites: Clinics and community-based clinics (urban) ≥50 years	Cross-sectional study Patients presenting for primary care appointments *n* = 345	White vs. minority Site 1 (*n* = 152): White 73% Site 2 (*n* = 193): White 57%	Majority accepted screening After exposing various potential risks and benefits, more accepted screening. No difference between white and minority
Fowler et al. (2012) [30]	U.S. Community-based clinics (urban) ≥65 years	Cross-sectional study Patients with no dementia receiving primary care *n* = 554	White 41.5%, African Americans 56.5% Other 1.4%	Majority willing to screen; 12.7% screened positive Refusal rates did not vary with ethnicity, education, other SES. Odds of refusal higher in older age groups
Fowler et al. (2015) [31]	U.S. 2 sites Clinic (urban) and community-based clinics (urban and suburban) ≥65 years	Cross-sectional study Patients with no dementia receiving primary care *n* = 400	Site 1 (*n* = 278): White 78.1%, African American 20.9%, Other 1.1% Site 2 (*n* = 122): White 96.7%, African American 2.5%, Other 0.8% English	Site 1: No difference in acceptance and refusal between White and African Americas Site 2: only one African American participant No differences in refusal between two sites Perceptions about the benefits of screening associated with acceptance of screening No effect of sociodemographic data except education predicted acceptance
Savva et al. (2015) [32]	U.S. Nationwide ≥71 years	Cross-sectional study Patients with dementia Data source: ADAMS substudy from HRS 2000–2002 waves *n* = 307	White 73%, Non-White 27% English or Spanish	121 informants reported prior diagnosis Grater CDC rate associated with prior diagnosis Race or nursing home residency no link with prior diagnosis Aged <90 years or married women associated with prior diagnosis. ¾ undiagnosed have mild dementia
Casado et al. (2017) [33]	U.S. Community (community outreach, local business sites, flyers, newsletters, social media) ≥40 years	Cross-sectional survey Adults *n* = 234	Korean Americans English or Korean	20.7% reported having experience with caring for someone with AD. Attitude scores were slightly more positive toward AD specialists (mean = 55.92 ± 7.40) than toward PCPs (mean = 54.24 ± 9.82).
Amjad et al. (2018) [34]	U.S. Nationwide ≥65 years	Cross-sectional observational study Patients with probable dementia or proxy Data source: NHATS *n* = 585	Non-Hispanic White, non-Hispanic Black, Hispanic or other non-Hispanic (Asian, Pacific Islander, and Native American)	39.5% undiagnosed Among diagnosed, 31% of those persons or their proxies were unaware of diagnosis. Undiagnosed persons likely to be non-White and lower education. But OR was statistically significant only for Hispanic/other non-White race Majority of older adults with dementia either undiagnosed or unaware of the diagnosis
Harrawood et al. (2018) [35]	U.S. 3 sites: clinic (urban) and community-based clinics (urban and suburban) ≥65 years	Cross-sectional study Patients with no dementia receiving primary care *n* = 954	African American 42.4% (*n* = 317): Site 1 (*n* = 280), Site 2 (*n* = 35), Site 3 (*n* = 2)	21.6% refused screening 78.4% agreed to be screened 10.2% screened positive: 11.7% African American; 9.0% White and other Older age (>75 years) low education, and perceived problem with memory associated with screening positive but no effect from race and research sites.
Gianattasio et al. (2019) [7]	U.S. Nationwide ≥70 years	Longitudinal study Patients with dementia Data source: HRS biannual interviews with participants or proxy linked with Medicare claims *n* = 4647–5201 (2000 to 2010, 6 observations)	Non-Hispanic White: 91–93%, Non-Hispanic Black: 7–9% English or Spanish	Whites were “correctly diagnosed” Blacks were “underdiagnosed” Black had double the risk of underdiagnosed compare with White at all 6 waves Risk of over diagnosed increased over time in both groups
Park et al. (2020) [36]	U.S. Community (urban, suburban, and rural) >50 years	Cross-sectional study Individuals with no dementia *n* = 1043	White 82.7%, Black/African American 11.6%, Hispanic 1.2%, Asian 0.6%, Native Hawaiian/Pacific Islander 0.3%, American Indian 0.8% English	In terms of demographic difference, female and participants with long-term care insurance have greater intention to screen but no mention about the effect of race. Younger age, higher level of perceived barriers, perceived benefit, higher social support and self-efficacy associated with increased intention
Lin et al. (2021) [37]	U.S. Nationwide ≥70 years	Prospective cohort study Patients with probable dementia Data source: HRS 2000–2014 linked with Medicare and Medicaid *n* = 3966	Non-Hispanic White 80.8%, non-Hispanic Black 11.9%, Hispanic 7.3% English or Spanish	A higher proportion of Blacks and Hispanics had a missed/delayed clinical dementia diagnosis compared with White (46%, s = 54% vs. 41%) Blacks and Hispanics had a poorer cognitive function and more functional limitations than White when received dementia diagnosis. Estimated mean delay: Blacks: 34.6 months; Hispanic 43.8 months; White: 31.2 months
Tsoy et al. (2021) [38]	U.S. Statewide	Retrospective cross-sectional study Patients with no prior dementia or MCI Data source: California CMS claims 2013–2015 *n* = 10,472	White 74.6%, Black 3.9%, Hispanic 12.0%, Asian 9.5%	Incident MCI diagnosis 23.3% White, 18.28% Black, 12.3% Asian, 15.8% Hispanic Timeliness of diagnosis Asian, Blacks, and Hispanic less likely to receive an incident diagnosis of MCI vs. dementia than White Estimated mean marginal effects of race/ethnicity on incident diagnosis of MCI were −11.0% for Asian, −6.6% for Hispanic, and −5.6% for Black
Wiese et al. (2019) [39]	U.S. Local service organizations, physician offices, church councils, senior center (rural)	Mixed methods Stakeholders -Social workers, healthcare administrators, nurses, nurses’ aides, physician, ministers, clerical worker, kitchen aids, farmworkers, auto mechanic, church worker *n* = 21	Non-Hispanic White (*n* = 5): professionals 4, layperson 1 African American (*n* = 11): professionals 9, laypersons 2 Afro-Caribbean (*n* = 2): professional 1, layperson 1 Hispanic American (*n* = 2): professional 1, layperson 1 English	81%: willing to screening annually if they developed memory problems or AD 85% of those previously screened would want to know if they were at higher risk of AD.
Wiese et al. (2021) [40]	U.S. Local city hall, senior centers, healthcare clinics, faith-based organizations (rural)	Mixed methods Stakeholders -Senior center administrators, senior center volunteer staffs, health clinic administrators, law enforcement officers, emergency medical technicians, physicians, nurse practitioners, nurses, paid caregivers, family caregivers, residents *n* = 22	White (*n* = 21), African American (*n* = 1) English	100%: willing to screening 82%: agreeable to blood testing 86%: agreeable to pictures of head or brain to detect dementia All would want their provider to screen them annually for memory problems
Williams et al. (2010) [41]	U.S. Churches, senior centers, health fair (announcements and flyers)	Mixed method Open-ended questions A part memory screening study of 793 community dwelling older adults *n* = 119	African American (*n* = 26) Afro-Caribbean (*n* = 31) European American (*n* = 29) Hispanic American (*n* = 33) English or Spanish	More African Americans recruited from churches than Hispanic and European American 89% valued the screening 92% would recommend screening to others 39% would seek professional help if they screened positive. More Hispanic Americans (70%) planned to seek help than did than European Americans (35%), African Americans (31%), or Afro-Caribbean (16%).
Hinton et al. (2004) [42]	U.S. Community (urban) (Physician referrals, Alzheimer’s Association, newspaper advertisements, etc.) Caregiver to patient ≥50 years	Qualitative Study In-depth interview A part of Survey Study: 33% of 117 family caregivers to community dwelling dementia patients -Wife, daughters, sons, others *n* = 39	African American (*n* = 10) Chinese American (*n* = 14) Anglo European-American (*n* = 15) English or three Chinese dialects (Mandarin, Cantonese, and Toisanese)	Help-seeking was most often initiated by family members or formal care providers Lack of a final diagnosis: more commonly reported by Chinese Americans compared with Anglos and African Americans Fragmentation in the referral process was common across all groups. Four general types of pathways to diagnosis: Smooth pathways/fragmented pathways/crisis pathways/dead-ended pathways
Hugh et al. (2009) [43]	U.S. Community (urban and rural) (A dementia outreach partnership)	Qualitative study Face-to-face semi-structured interview Health belief model Family caregivers of dementia patients -Daughters, spouses, sons, siblings *n* = 17	African American	Not knowledgeable about AD prior to their family diagnosed Knew that there is no known cure and expected a continued decline Almost half attributed a change in cognition was normal, age-related memory loss Some caregivers received support or resistance from other family member A supportive social network facilitated a diagnosis. Perplexing behavior and an increasing loss of ability are seen as cues to action
Leung et al. (2011) [44]	Canada Community and clinic (urban) (Alzheimer’s Society, posters) Patients: >55 years	Qualitative study Semi-structured interview Dyads of patients with dementia and family caregivers -Caregivers: wives, daughter, son-in-law, husband *n* = 6 dyads (7 caregivers)	Anglo-Canadian English	Symptom recognition to a dementia diagnosis 2–4 years Demented patients noticing memory difficulties earlier than careers but perceived as ambiguous and normalized or attributed to current health problem Diagnosis process was multiple visits and interactions with health professionals, obtained as more severe cognitive deficit emerged
Koehn et al. (2012) [45]	Canada Community (urban) (Chinese Resource Center of Alzheimer’s Society)	Qualitative study Semi-structured interview A Help-seeking Model Dyads of patients with probable dementia and their careers -Caregivers: wives, husband, daughter *n* = 10 dyads	Chinese Canadian Cantonese or Mandarin	The average pre-diagnosis interval: 1.5 years Caregivers and patients reported a diversity of experiences regarding the early symptoms of the patients’ cognitive deficit. Normalized of early symptoms Decision to seek care was made by family member, either spouse or consulted with adult children Two diagnosed done during acute care admission The role of family caregivers was more influenced by structural factors than by traditional Chinese cultural norms about family responsibilities and filial piety. 60% of the dyads experienced delays in diagnosis because Chinese family doctors dismissed the caregivers’ appraisals of the patients’ symptoms. Gender-based power imbalance between female family caregivers and male Chinese Canadian physicians
McCleary et al. (2012) [46]	Canada Community (urban) (Adult daycare center and flyers to community health center, local Alzheimer’s Society) Patient: >70 years	Descriptive qualitative study Semi-structured interview Dyads of patients with dementia and either one or two of their family careers -Caregivers: wives, daughters, daughter-in-law, husband, son, son-in-law *n* = 6 dyads	South Asian-Canadian English, Hindi, or Tamil	Early signs of dementia were seen as normal that are related to the aging process or patients’ personality characteristics. Seek attention when dementia symptoms were worsened after episodes Health seeking was delayed up to four years, even with significant dementia symptoms Safety concerns, new symptoms, treatment for other health problem influenced the recognition of a health problem
Garcia et al. (2013) [37]	Canada Clinic setting (a memory disorder clinic) Patients: >60 years	Qualitative study Semi-structured interview Dyads of patients with dementia and family or friends -Caregivers: spouses, daughters *n* = 7 dyads	French-speaking Canadian French	Estimated first suspicion of a problem to an official diagnosis: 1–7 years Not easy to identify signs and symptoms Lack of knowledge about the importance of the changes they were experiencing. No single symptoms sufficient to alert participants Preferentially sought from francophone Recognition to consultation with family physician from 4 months to 6 years. All final diagnoses were made by specialists, but family physicians clearly suspected dementia Variety of reasons for the delay.

Abbreviations: AD, Alzheimer’s disease; ADAMS, Aging, Demographics and Memory Study; HRS, Health and Retirement Study; CDC, Centers for Disease Control and Prevention; PCP, primary care provider; NHATS, National Health and Aging Trends Study; OR, odds ratio; MCI, mild cognitive impairment; CMS, Centers for Medicare & Medicaid.

**Table 2 ijerph-19-08865-t002:** Definitions and Measures of ADRD Screening Behavior.

Dimensions	Definitions	Findings: Empirical Statements	Measures/ Example Items
Noticing symptoms	Noticing first signs and symptoms	“I heard the word, but I did not pay much attention to it” Usually normalized or attribute to other health problems. (Hugh et al. 2009) [43]	“The first ADRD symptoms were observed” (Clark et al. 2005) [28]
Recognizing a problem	Recognize the signs and symptoms as problems	Multiples signs and symptoms, more cognitive and behavior changes, symptoms getting worse and increasing loss of ability. (McCleary et al. 2013) [46]	“Caregiver’s recognition that a problem existed” (Clark et al. 2005) [28]
Accepting Screen	Acknowledging that there is a problem that needs to change	Accumulation of subtle changes including issues with hygiene, finance, or safety in combination with forgetfulness; Consider harm and benefit of screening. (Garcia et al. 2013) [47]	Modified SAPH ^*^ “I would like to know if I have a problem with my memory that may indicate that I’m developing dementia” (Holsinger et al. 2011) [29]
Intending Screen	Intention or willingness undergo screen	“No critical event either physical or cognitive symptoms to trigger their desire to seek care” “Concerns safety is cue” (Leung et al. 2011) [44]	“Plan to screen for AD at some point in life” “Plan to screen for AD in the next year” “Plan to screen for AD after the participant reaches a certain age” “Plan to screen for AD in the presence of symptoms for AD” (Park et al. 2020) [36]
Action	Taking actions to assess the symptoms	Diagnosis process was multiple visits and interactions with health professionals. (Leung et al. 2011) [44]	Ever participated in screen procedures “Patients completed the MMSE, CSI-D, or TICS” (Harrawood et al. 2018) [35]
Integrating with time	Timeliness of receiving a clinical diagnosis; Delays in diagnosis of ADRD	Delays due to not only the trajectory of the disease and patients’ personality but also to the types of caregivers. (Koehn et al. 2012) [45] There was no single common pathway from recognition to action. (McCleary et al. 2013) [46]	“The first observed ADRD symptoms to first physician visit” (Clark et al. 2005) [28]

Abbreviations: ADRD, Alzheimer’s disease and related dementias; SAPH, Dementia Screening and Perceived Harms Questionnaire; PRISM-PC, Perceptions Regarding Investigational Screening for Memory in Primary Care; AD, Alzheimer’s disease; CSI-D, Community Screening Instrument for Dementia; TICS, Telephone Instrument for Cognitive Screening. * SAPH has become the PRISM-PC (Acceptance subscale: 2 dimensions, 6 items).

**Table 3 ijerph-19-08865-t003:** ADRD screening behaviors by race/ethnicity.

Non-Hispanic Black vs. White				
Subgroups	Findings			
	Reference	Outcome	Value	Reference Group (Non-Hispanic White)
Non-Hispanic Black	Fowler et al. (2015) [31]	Accepted screening		
		Urban hospital	60.3%	63.1%
		Network of urban and suburban hospitals and outpatient care centers	66.3%	63.6%
	Folwer et al. (2012) [30]	Undergo screening	89.5%	90.4%
	Harrawood et al. (2018) [35]	Screened positive for dementia	11.7%	9% ^†^
	Lin et al., (2021) [37]	Dementia diagnosis without delay	54.5%	59.3%
		Missed or delayed dementia diagnosis	45.5%	40.8%
	Amjad et al. (2018) [34]	Undiagnosed vs. diagnosed dementia	Adjusted OR 1.26 ^‡^	-
		Unaware vs. aware of dementia diagnosis	Adjusted OR 0.73 ^‡^	
	Gianattasio et al. (2019) [7]	Underdiagnosed	Adjusted PR 1.35–2.33	-
	Hinton et al. (2004) [42]	Lack of a final diagnosis	20%	7%
Latino/Hispanic vs. White				
Subgroups	Findings			
	Reference	Outcome	Value	Reference group (Non-Hispanic White)
Hispanic	Lin et al. (2021) [37]	Dementia diagnosis without delay	45.8%	59.3%
		Missed or delayed dementia diagnosis	54.2%	40.8%
	Amjad et al. (2018) [34]	Undiagnosed vs. diagnosed dementia	Adjusted OR 2.48	-
		Unaware vs. aware of dementia diagnosis	Adjusted OR 0.87 ^‡^	-
Asian vs. White				
Subgroup	Findings			
	Reference	Outcome	Value	Reference group (Non-Hispanic White)
Chinese	Hinton et al. (2004) [42]	Lack of a final diagnosis	43%	7%
Other vs. White				
Subgroup	Findings			
	Reference	Outcome	Value	Reference group (Non-Hispanic White)
Other	Fowler et al. (2015) [31]	Accepted screening		
		Urban hospital	33.3%	63.1%
		Network of urban and suburban hospitals and outpatient care centers	100%	63.6%
	Fowler et al. (2012) [30]	Undergo screening	87.5%	90.4%
Non-White	Savva et al. (2015) [32]	Prior diagnosis of dementia (weighted)	48%	41.1%

^†^ “Other” racial groups are also included in the reference group. ^‡^ No statistical significance. Abbreviations: OR, odds ratio; PR, prevalence ratio.

## Data Availability

Not applicable.

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
