# Peer review of "Examine Race/Ethnicity Disparities in Perception, Intention, and Screening of Dementia in a Community Setting: Scoping Review"

_ijerph, 2022, doi:10.3390/ijerph19148865_

Round 1

Reviewer 1 Report

First, I wish to thank the authors for choosing to study racial differences in both the perception of symptoms of Alzheimer's disease and other causes of cognitive impairment and in the diagnostic strategies employed. However, I have identified a number of aspects that I believe need to be improved before publication can be considered:
1) I believe that the concept itself is erroneous. At present, the diagnosis of Alzheimer's disease (AD) and/or other causes of cognitive impairment at the dementia stage is a mistake (is to late). The focus should be on differences in the diagnosis of mild cognitive impairment (MCI). The response to symptomatic treatmentes currently available for AD is much higher in MCI stage or early dementia, and the inclusion in clinical trials is more probable in case of MCI. 

Currently, a treatment with possible (is not so clear) modifying effect in AD has been approved (Aducanumab) by FDA but not by EMA. There is not mention at all to this treatment.

2) When talking about dementias related to Alzheimer's disease, what do you mean  copathologies or causes that raise differential diagnosis with them? It is not clear. Explain. Also describe possible copathologies and/or causes of differential diagnosis.
It would also be important to point out Alzheimer's disease as the first cause of dementia. It is not clear.
There is also no mention of screening strategies for mild cognitive impairment that have changed radically in recent years (especially at the stage of mild cognitive impairment).
3) It is highly likely that the level of schooling differs radically between patients and their caregivers according to racial group. There is no mention in this regard which I believe is crucial: the lower the educational level of the patient the higher the probability of late diagnosis and the higher the probability of not identifying the first symptoms early.
4) Access to universal health care also differs according to race in countries such as the United States. In order to correctly evaluate the racial impact on the delay in syndromic and etiological diagnosis (this is not specified either and it would be expected that there would be differences between the two), I believe it would be better to consider the effect on a population with universal access to the public health system, as is the case in European countries.

5) Trying to assess the sociofamiliar impact of the diagnosis of Alzheimer's disease or related disorders could be also of interest. I believe that the access to certain social resources could be clearly delayed depending of the race.
5) Table 1 is difficult to read. Creating categories that divide the studies that provide evidence in screening by health professionals, by caregivers... would make it more attractive. Right now it is not. 
6) Adding some informative panel that collects the key ideas and/or a figure that summarizes them would help the reading of this work.
7) There are multiple statements without proper references.
8) Table 2 is of little interest as written. I understand the idea but it should be rethought: dimension and definition are ok although there are some definitions that could be improved, the third column for me does not contribute anything and the last one could be redrafted.
9) Including recommendations/outlines of strategies to reduce the differences identified would also make the work much more interesting.

I am not continuing with any more suggestions for change until the ones noted here are resolved.

Author Response

Dear Reviewer, 

Thank you so very much for your comments.  We have diligently reviewed each comment and have addressed the concerns and suggestions made. Below you will find detailed responses to each comment.

We are excited about the potential to have this manuscript published by the International Journal of Environmental Research and Public Health. We welcome any further suggestions and/or comments.

Sincerely,

All authors

Reviewer’s comments

First, I wish to thank the authors for choosing to study racial differences in both the perception of symptoms of Alzheimer's disease and other causes of cognitive impairment and in the diagnostic strategies employed. However, I have identified a number of aspects that I believe need to be improved before publication can be considered:

1) I believe that the concept itself is erroneous. At present, the diagnosis of Alzheimer's disease (AD) and/or other causes of cognitive impairment at the dementia stage is a mistake (is to late). The focus should be on differences in the diagnosis of mild cognitive impairment (MCI). The response to symptomatic treatments currently available for AD is much higher in MCI stage or early dementia, and the inclusion in clinical trials is more probable in case of MCI.  Currently, a treatment with possible (is not so clear) modifying effect in AD has been approved (Aducanumab) by FDA but not by EMA. There is not mention at all to this treatment.

Response to the Reviewer

We appreciate your comments focusing on the differences in the diagnosis of MCI and a treatment with a possible modifying effect in AD. Though this is a very important topic to be studied, the aim of this paper is to investigate ADRD screening seeking behavior.

Reviewer’s comments

2) When talking about dementias related to Alzheimer's disease, what do you mean  copathologies or causes that raise differential diagnosis with them? It is not clear. Explain. Also describe possible copathologies and/or causes of differential diagnosis. It would also be important to point out Alzheimer's disease as the first cause of dementia. It is not clear. There is also no mention of screening strategies for mild cognitive impairment that have changed radically in recent years (especially at the stage of mild cognitive impairment).

Response to the Reviewer

Regarding the need to explain co-pathophysiological mechanisms for difference diagnosis:  Thank you. Though it is very important to study co-pathologies and causes to understand the different diagnoses of AD and MCI, for this paper, we only reviewed studies that addressed ADRD screening/diagnosing behavior but do not intend to describe possible co-pathologies and/or other causes of different diagnoses.  

Reviewer’s comments

3) It is highly likely that the level of schooling differs radically between patients and their caregivers according to racial group. There is no mention in this regard which I believe is crucial: the lower the educational level of the patient the higher the probability of late diagnosis and the higher the probability of not identifying the first symptoms early.

Response to the Reviewer

Thank you for your comment: Sociodemographic factors related to ADRD screening seeking behaviors: educational level, health care access, etc.: Thank you for your suggestions.  Yes, we do also think other sociodemographic factors might influence ADRD screening seeking behavior, however, our major interest is in examining the impact of race/ethnicity on ADRD screening seeking behavior. As we stated earlier, the purpose of this study is to exam how and what “race/ethnicity” a socio-demographic factor, affects ADRD screening/diagnosing seeking behavior. However, we have added more information in the section on strengths and limitations on how educational levels and health care access could function as moderating factors for ADRD screening behavior and further studies are necessary to determine it.

Reviewer’s comments

4) Access to universal health care also differs according to race in countries such as the United States. In order to correctly evaluate the racial impact on the delay in syndromic and etiological diagnosis (this is not specified either and it would be expected that there would be differences between the two), I believe it would be better to consider the effect on a population with universal access to the public health system, as is the case in European countries.

Response to the Reviewer

Thank you for your comment. Please see responses to item 3.

Reviewer’s comments

5) Trying to assess the sociofamiliar impact of the diagnosis of Alzheimer's disease or related disorders could be also of interest. I believe that the access to certain social resources could be clearly delayed depending on the race.

Response to the Reviewer

Thank you for your comment. Please see responses to item 3.

Reviewer’s comments

5-2) Table 1 is difficult to read. Creating categories that divide the studies that provide evidence in screening by health professionals, by caregivers... would make it more attractive. Right now it is not. 

Response to the Reviewer

Thank you for your comment. Yes, we do think that the category of screening by health professionals or by caregivers would be a very important topic to be studied, however, the main focus of this study is to understand the effect or impact of race/ethnicity on ADRD screening behavior.

Reviewer’s comments

6) Adding some informative panel that collects the key ideas and/or a figure that summarizes them would help the reading of this work.

Response to the Reviewer

As stated in the section of Material and Methods, we rigorously followed the PRISMA-ScR and O’Malley’s scoping review guidelines and provided thick descriptions of our protocol.

Reviewer’s comments

7) There are multiple statements without proper references.

Response to the Reviewer

Thank you for your comment.  We added more references to give supporting evidence for our arguments and statements.

Reviewer’s comments

8) Table 2 is of little interest as written. I understand the idea but it should be rethought: dimension and definition are ok although there are some definitions that could be improved, the third column for me does not contribute anything and the last one could be redrafted.

Response to the Reviewer

Thank you for your comment. Yes, as you noticed that ADRD screening behavior is a very difficult phenomenon to understand and to study. As you suggested, we rewrote it as:

As listed in Table 2, the construct ADRD screening behavior that we identified is multi-dimensional and is not a static phenomenon.  This ADRD screening behavior took place across five main dimensions: 1) noticing symptoms, 2) recognizing a problem, 3) accepting screening, 4) intending to screen, 5) action, and 6) integrating with time.  As for the dimension of integrating with time of ADRD screening seeking behavior, we included delays and timely screenings. Most studies focused on recognition of the problem 210 (n=6), acceptance or intention (n=4) to do screening of ADRD, and actual ADRD screening (n=4). The empirical statements for each dimension extracted from the included studies were summarized and compared in Table 2.

As for measurements, there is limited information assessing the multi-dimensional and time-bounded pheromone of ADRD seeking behavior.  In general, there is an absence of a commonly agreed upon approach to assess ADRD seeking behavior as shown in the Table 2.

Reviewer’s comments

9) Including recommendations/outlines of strategies to reduce the differences identified would also make the work much more interesting.

Response to the Reviewer, thank you for your comment, we added recommendations under the conclusion section. 

Reviewer 2 Report

Thanks for all efforts done to address this very important issue regarding the health disparities in dementia care and cognitive screening. 

Discrimination in health services and inequity in cognitive screening is underdiagnosed and we should pay more attention to address this problem.

I believe this article would be stepping forward for stakeholders and researchers to be aware about it.

Author Response

Thank you for acknowledging the significance of our work and your belief that this article would be an igniter for further inquiries and researchers, health care providers, health policy makers, and stakeholders will all together pay more attention to address the problems of ADRD screening seeking behaviors, especially, underserved and understudied ethnic-minority populations. 

Reviewer 3 Report

Lee and colleagues performed a systematic review on racial/ethnic disparities in misdiagnosis/underdiagnosis of dementia using PRISMA-ScR. Adequate search terms were used and to increase consistency among reviewers, three independent researchers screened all discovered studies and reached a consensus through post hoc discussions. The methods are generally well described. The only suggestions I have are content-wise:

The introduction touches on a very good argument regarding the fact that racial/ethnic minorities are usually more impaired and show more severe clinical symptoms when presented for clinical screening. Interestingly, these findings during life also translate to pathology studies, in which racial/ethnic minorities are much less represented in Brain bank donation programs (see e.g. Santos Alzheimers Dement. 2019). Timely detection of dementia is equally important as well as knowing the final neuropathological diagnosis. Although the topic doesn’t deal with pathology studies, I personally found this an interesting observation.

In the results, it is mentioned that involving physicians who speak the same ethnic language could improve timely diagnosis of dementia. This is an important consideration. However, this point isn’t addressed anymore in detail in the discussion (except for Limitations). Given that this review focused on studies performed in US (English) and/or Canada (English/French), the language barrier is considered one of the main barriers to ADRD screening behaviour in these monolingual countries. This observation might differ from countries where multiple languages are commonly spoken (e.g. in several European countries French is one of the languages (Belgium, Luxembourg, France, Swiss,  ..) and this makes that e.g. Northern African patients, who often have French as second language, can still be screened). Perhaps the authors could discuss this in more detail.

Author Response

The comments and feedback were very constructive and encourage changes which greatly strengthen our manuscript as you will notice. 

Yes, we believe health literacy is so important, especially among immigrant populations who often have significant language and health literacy difficulties in addition to cultural barriers to access the health care system and get relevant health information.

As you observed, only two studies reported the language matched between patients (or caregivers) and physicians and provided very limited information to address this important topic in detail.  So, we added more information about this topic in the section on strengths and limitations and the section of conclusion. We have tried to capture your points in our manuscript.  Thank you so much for all your valuable and constructive feedback which really strengthens our manuscript.

Round 2

Reviewer 1 Report

First of all, I would like to thank the authors for the modifications they have done following the suggestions made for the reviewers. I do still consider that the work could be improved and so, its potential relevance of the field increased but I am very aware that this would imply for a different design of the study (almost a new work). So, I going to recommend to evaluate the publication of the work at the present form, because I do believe that it could have an interest for clinical practice, but I encourage the authors to keep exploring and working in this topic.